# Human Serum Proteins and Susceptibility of *Acinetobacter baumannii* to Cefiderocol: Role of Iron Transport

**DOI:** 10.3390/biomedicines10030600

**Published:** 2022-03-03

**Authors:** Casin Le, Camila Pimentel, Fernando Pasteran, Marisel R. Tuttobene, Tomás Subils, Jenny Escalante, Brent Nishimura, Susana Arriaga, Aimee Carranza, Vyanka Mezcord, Alejandro J. Vila, Alejandra Corso, Luis A. Actis, Marcelo E. Tolmasky, Robert A. Bonomo, Maria Soledad Ramírez

**Affiliations:** 1Center for Applied Biotechnology Studies, Department of Biological Science, College of Natural Sciences and Mathematics, California State University Fullerton, Fullerton, CA 92831, USA; thanhle1998@csu.fullerton.edu (C.L.); camilapimentel99@csu.fullerton.edu (C.P.); jenni1@csu.fullerton.edu (J.E.); bnish-942@csu.fullerton.edu (B.N.); arriagasusie@gmail.com (S.A.); aimeecarranza1119@gmail.com (A.C.); mezcordvyanka@csu.fullerton.edu (V.M.); mtolmasky@fullerton.edu (M.E.T.); 2National/Regional Reference Laboratory for Antimicrobial Resistance (NRL), Servicio Antimicrobianos, Instituto Nacional de Enfermedades Infecciosas, ANLIS Dr. Carlos G. Malbrán, Buenos Aires C1282, Argentina; fpasteran@anlis.gov.ar (F.P.); acorso@anlis.gob.ar (A.C.); 3Área Biología Molecular, Facultad de Ciencias Bioquímicas y Farmacéuticas, Universidad Nacional de Rosario, Rosario S2000, Argentina; tuttobene@ibr-conicet.gov.ar; 4Instituto de Biología Molecular y Celular de Rosario (IBR, CONICET-UNR), Rosario S2000, Argentina; vila@ibr-conicet.gov.ar; 5Instituto de Procesos Biotecnológicos y Químicos de Rosario (IPROBYQ, CONICET-UNR), Rosario S2002, Argentina; subils@iprobyq-conicet.gob.ar; 6Área Biofísica, Facultad de Ciencias Bioquímicas y Farmacéuticas, Universidad Nacional de Rosario, Rosario S2000, Argentina; 7Department of Microbiology, Miami University, Oxford, OH 45056, USA; actisla@miamioh.edu; 8Departments of Medicine, Pharmacology, Molecular Biology and Microbiology, Biochemistry, Proteomics and Bioinformatics, School of Medicine, Case Western Reserve University, Cleveland, OH 44106, USA; 9Research Service and GRECC, Louis Stokes Cleveland Department of Veterans Affairs Medical Center, Cleveland, OH 44106, USA; 10CWRU-Cleveland VAMC Center for Antimicrobial Resistance and Epidemiology (Case VA CARES), Cleveland, OH 44106, USA

**Keywords:** *Acinetobacter baumannii*, cefiderocol, human fluids, iron, human serum albumin

## Abstract

Cefiderocol, a recently introduced antibiotic, has a chemical structure that includes a cephalosporin that targets cell wall synthesis and a chlorocatechol siderophore moiety that facilitates cell penetration by active iron transporters. Analysis of the effect that human serum, human serum albumin, and human pleural fluid had on growing *Acinetobacter baumannii* showed that genes related to iron uptake were down-regulated. At the same time, β-lactamase genes were expressed at higher levels. The minimum inhibitory concentrations of this antimicrobial in *A. baumannii* cells growing in the presence of human serum, human serum albumin, or human pleural fluid were higher than those measured when these fluids were absent from the culture medium. These results correlate with increased expression levels of β-lactamase genes and the down-regulation of iron uptake-related genes in cultures containing human serum, human serum albumin, or human pleural fluid. These modifications in gene expression could explain the less-than-ideal clinical response observed in patients with pulmonary or bloodstream *A. baumannii* infections. The exposure of the infecting cells to the host’s fluids could cause reduced cefiderocol transport capabilities and increased resistance to β-lactams. The regulation of genes that could impact the *A. baumannii* susceptibility to cefiderocol, or other antibacterials, is an understudied phenomenon that merits further investigation.

## 1. Introduction

Carbapenem-resistant *Acinetobacter baumannii* (CRAB), one of the most feared pathogens in healthcare settings, has been categorized by the Centers for Disease Control and Prevention (CDC) as an “urgent threat” [1,2,3,4,5]. As a consequence of *A. baumannii*’s ability to develop multidrug resistance (MDR), treatment strategies have become extremely limited, with only a few active antibiotics in existence [6,7,8].

The increasing number of CRAB infections translates into alarmingly high morbidity and mortality [2,9,10,11]. Consequently, numerous efforts are focusing on finding novel treatment options [12,13,14,15,16,17,18,19]. Cefiderocol (CFDC) was approved by the Food and Drug Administration of the United States (FDA) in November 2019 to treat nosocomial pneumonia and urinary tract infections (Available online: https://www.accessdata.fda.gov/drugsatfda_docs/label/2019/209445s000lbl.pdf (accessed on 14 February 2022)) [20]. CFDC is a hybrid molecule that consists of a cephalosporin component that targets cell wall synthesis and a catechol siderophore moiety that allows cell penetration by active ferric-siderophore transporters [21,22,23,24,25]. This novel synthetic compound uses a “Trojan horse” strategy to improve antibiotic penetration and reach a high concentration at the target site [22,23]. This strategy allows the drug to enter the bacterial cells using active iron transporters. In *A. baumannii*, the most well-characterized iron transporters are *bauA*, *bfnH*, and *fbsN*, which are coupled to a TonB system to translocate the siderophore–iron complex into the periplasm [26]. However, up to 21 putative TonB-dependent outer membrane transporter genes have been identified or predicted in *A. baumannii* genomes, most often associated with putative or confirmed ferric-siderophore and heme uptake genes [27]. The *bauA* gene, which encodes for the major outer membrane receptor of ferric–acinetobactin complexes, was extensively studied in *A. baumannii* and was shown to be highly conserved [28,29,30,31]. In addition, PiuA and PirA, and also TonB-dependent receptors, were shown to mediate the uptake of siderophore–iron complexes [32]. These receptors have been associated with the cellular entry of CFDC in Gram-negative pathogens [25,33,34]. CFDC′s primary targets are cell wall synthesizing proteins associated with β-lactam activity, such as the penicillin-binding proteins 1 and 3 (PBP1 and PBP3) [35]. Randomized clinical trials, and other small case series studies, showed CFDC as a promising alternative to treat carbapenemase-producing pathogens and other less frequent non-fermenters, such as *Stenotrophomonas maltophilia*, *Burkholderia* spp., and *Achromobacter* spp [7,22,36,37,38,39,40,41].

Simner et al. explored possible resistance mechanisms to CFDC in 14 isolates of carbapenem-resistant Enterobacterales that exhibited baseline CFDC resistance (minimum inhibitory concentration (MICs) >/= 4 mg/L) [42]. They identified heterogeneous mechanisms that include the combination of β-lactamase production and permeability defects contributing to elevated CFDC MIC. Particularly, some of the mechanisms identified were the presence of New Delhi metallo-β-lactamase (NDM) carbapenemases, the occurrence of outer membrane porin mutations, and the presence of efflux transporters among the isolates evaluated. In addition, mutations in the *baeS* gene, responsible for encoding a sensor kinase protein of the two-component BaeSR and mutations in *tonB3*, *exbD*, *envZ*, or *ompR*, were identified among the clinical isolates that exhibited elevated CFDC MICs. The down-regulation of specific iron transport receptors in *Pseudomonas aeruginosa*, named FecIRA, is correlated with resistance to CFDC [43]. Similarly, in *A. baumannii*, CFDC resistance was associated with the reduced expression of the TonB-dependent receptor gene *pirA* as well as mutations in the penicillin-binding proteins 3 (PBP_3_). The presence of acquired β-lactamases, such as PER, may have also contributed to resistance in some strains [44,45]. So far, studies have not linked β-lactamase activity, nor levels of expression of PBPs, to CFDC resistance.

*A. baumannii* senses components of human fluids and responds by modifying its transcriptional and phenotypic profiles [46,47,48,49,50]. Human serum albumin (HSA), as well as human pleural fluid (HPF), modulates the expression of genes associated with iron uptake systems, biofilm formation, antibiotic resistance, and DNA acquisition, among others [47,51,52]. In previous studies, we observed that CRAB AB5075 genes associated with iron uptake systems were down-regulated when exposed to HPF and 0.2% HSA [47], while genes associated with β-lactam resistance were up-regulated in the presence of physiological concentrations of HSA and human serum (HS) [51,52]. Understanding the modification in the expression of iron uptake systems and β-lactam resistance genes in *A. baumannii* exposed to biological fluids could allow future strategies to protect antibiotics from a possible deleterious in vivo efficacy. In this work, we describe the effect of HS, HSA, and HPF on representative CRAB strains with different genetic backgrounds, focusing our analysis on the level of expression of iron uptake TonB-dependent receptors and β-lactam resistance genes.

## 2. Materials and Methods

### 2.1. Bacterial Strains

The model multidrug-resistant *A. baumannii* strain AB5075 was used [47,52,53]. In addition, *A. baumannii* AMA16 and AB0057 strains were also used [54,55,56,57]. The AB5075 strain contains the carbapenem resistance genes *bla*_OXA-23_, *bla*_GES-14_, and *bla*_ADC_. The AMA16 strain presents *bla*_NDM-1_, *bla*_PER-7_, and IS*Aba125* resistance genes. The AB0057 strain harbors the genes *bla*_TEM-1_, *bla*_OXA-23_, and *bla*_ADC_. This strain was isolated in the Walter Reed Army Medical Center, Bethesda, MD 20814, USA [58]. In addition, all the strains presented conserved resistance genes such as *bla*_OXA-51-like_, *carO*, *pbp1*, and *pbp3*. Nineteen additional CRAB strains, belonging to different clonal complexes, were included to performed CFDC susceptibility assays [54,55].

### 2.2. RNA Extraction and Quantitative Reverse Transcription Polymerase Chain Reaction (qRT-PCR)

Overnight cultures of *A. baumannii* strains were diluted 1:10 in fresh LB medium, LB medium containing 3.5% HSA (physiological concentration), containing 4% HPF, or 100% pooled normal HS, and incubated with agitation for 5 h at 37 °C. Pure HSA (Sigma-Aldrich, St. Louis, MO, USA) and pooled normal human serum from a certified vendor (Innovative Research Inc, Novi, MI, USA) were used in the cultures. RNA was extracted from each strain using the Direct-zol RNA Kit (Zymo Research, Irvine, CA, USA), following the manufacturer’s instructions. Total RNA extractions were performed in three biological replicates for each condition. The integrity of the RNA samples was verified by agarose gel electrophoresis. The extracted and DNase-treated RNA was used to synthesize cDNA using the manufacturer’s protocol provided within the iScriptTM Reverse Transcription Supermix for qPCR (Bio-Rad, Hercules, CA, USA). The cDNA concentrations were adjusted to 50 ng/µL and qPCR was conducted using the qPCRBIO SyGreen Blue Mix Lo-ROX, following the manufacturer’s protocol (PCR Biosystems, Wayne, PA, USA). At least three biological replicates of cDNA were used in triplets and were run using the CFX96 TouchTM Real-Time PCR Detection System (Bio-Rad, Hercules, CA, USA). Transcriptional levels of each sample were normalized to the transcriptional level of *rpoB*. The relative quantification of gene expression was performed using the comparative threshold method 2^−ΔΔCt^. The ratios obtained after normalization were expressed as folds of change compared with cDNA samples isolated from the bacterial cultures in the LB. Asterisks indicate significant differences as determined by ANOVA, followed by Tukey’s multiple comparison test (*p* < 0.05), using GraphPad Prism (GraphPad software, San Diego, CA, USA).

### 2.3. Antimicrobial Susceptibility Testing

Antibiotic susceptibility assays were performed following the procedures recommended by the Clinical and Laboratory Standards Institute (CLSI), Malvern, PA, USA [59]. After OD adjustment, 100 µL of cells grown in fresh LB medium, LB medium containing 3.5% HSA, 4% HPF, or 100% HS were inoculated on Mueller–Hinton agar plates, as previously described [47,48,50]. Antimicrobial commercial E-strips (Liofilchem S.r.l., 64026 Roseto degli Abruzzi, Italy) for CFDC was used. Mueller–Hinton agar plates were incubated at 37 °C for 18 h. CLSI breakpoints were used for interpretation [60].

## 3. Results

### 3.1. Changes in the Expression Levels of Genes Associated with Iron Uptake Systems in the Presence of Human Fluids

Building upon on our previous observations on the AB5075 strain, we decided to further study the role of HS, HSA, and HPF on CRAB AB0057 and AMA16 strains. The selected strains belong to different clonal complexes and harbor clinically important β-lactamases, such as oxacillinases (OXA-23) and NDM-1, respectively [54,61], which are the most prevalent carbapenemases in *A. baumannii* and Enterobacterales, respectively [4,62,63]. Quantitative RT-PCR (qRT-PCR) assays, to evaluate the expression of the iron uptake TonB-dependent receptor *pirA*, *piuA*, and *bauA*, were carried out using total RNA extracted from cells cultured in lysogeny broth (LB), LB supplemented with 4% HPF and 3.5% HSA, and cells cultured in 100% HS, as previously described [47]. In addition, the analysis of the expression levels of other iron uptake-associated genes, such as *tonB_3_*, *basE*, *feoA*, *fhuE1*, and *fhuE2*, involved in transport and biosynthesis [64,65,66,67,68], was also performed.

In the CRAB AMA16, the expression of the iron transport-related genes *bauA*, *pirA*, and *piuA* was reduced in all tested conditions evaluated (Figure 1A and Appendix A). Similar results were observed for the CRAB AB0057 strain. A reduction in the expression of *bauA*, *pirA*, and *piuA* in the tested condition was also seen (Figure 1B and Appendix A). In addition, when the hypervirulent *A. baumannii* AB5075 strain was exposed to 3.5% HSA and the HS, the expression levels of the genes encoding for the above-mentioned TonB-receptor genes were down-regulated (Figure 1C and Appendix A). Pimentel et al. previously showed that the levels of expression of AB5075 iron-related genes were down-regulated in cells cultured in iron-rich media [47]; this finding is in agreement with the level of expression of *basE* reported in this work (Appendix A).

Analysis of available RNA-seq data of AB5075 cultivated in LB vs. 3.5% HSA showed that *fhuE1, fhuE2, pirA*, and *piuA* genes [47,68] were also down-regulated in the presence of HPF (Appendix A). Moreover, the transcriptional levels of *fhuE1*, *fhuE2*, *tonB*, and *feoA* genes were further analyzed by qRT-PCR for the three strains (AMA16, AB0057, and AB5075) in the three tested conditions (HPF, HSA, and HS), showing a decrease in the level of expression in all of the strains in all of the tested conditions (Appendix A), with the exception of an increase in the expression of *feoA* for AMA16 exposed to HPF (Appendix A).

### 3.2. Modulation of the Expression of β-Lactam Resistance Genes by Human Fluids

We previously observed that genes associated with β-lactam resistance, such as *bla*_OXA-51-like_, *bla*_OXA-23_, *pbp1*, and *pbp3*, were up-regulated in *A. baumannii* AB5075, AMA16, and AB0057 when cultured in the presence of 3.5% HSA (51, 52). To further examine if changes in genes associated with β-lactam resistance were affected by HS or HPF in the CRAB strains, qRT-PCR was used to assess the gene expression of *bla*_OXA-51-like_, *bla*_OXA-23_, *bla*_NDM-1_, *pbp1*, *pbp3*, and *carO* in cultures of the strains AMA16 and AB0057 containing HS or HPF.

This transcriptional analysis revealed a down-regulation, with a log_2_ fold change of -2.84 and −1.62, in expression levels of *bla*_NDM-1_ in *A. baumannii* AMA16 cultured in the presence of HS or HPF, respectively (Figure 2A and Appendix A). In this strain, the levels of expression of ISA*ba125*, *bla*_PER-7_, *pbp1*, and *pbp3* were up-regulated in the presence of HPF (Appendix A). HS also induced the up-regulation of ISA*ba125* (Figure 2A). In the case of the AB0057 strain, we observed an increase in the expression of *bla*_OXA-23_, *bla*_OXA-51_, *carO*, *bla*_ADC_, *pbp1*, and *pbp3* in cultures containing HPF. In cultures containing HS, the former four genes were up-regulated (Figure 2B and Appendix A). Assessment of the changes in β-lactam resistance-associated *A. baumannii* AB5075 genes in the presence of HPF showed the down-regulation of the *carO* and *bla*_GES-14_ genes (Figure 2C). The *carO* gene encodes for an outer membrane protein channel that allows the permeation of imipenem in *A. baumannii* [69]. The marked down-regulation of *carO* (more than 4 Log2 fold change, Appendix A) could be contributing to limited CDFC entrance; however, further analysis is needed to establish this. On the other hand, the expression levels of *bla*_OXA-51_, *pbp1*, *pbp3*, and *bla*_ADC_ were increased (Figure 2C).

The impact of human fluids in CFDC activity was further studied using a panel of twenty CRAB strains, including AB5075, AMA16, and AB0057. Cells cultured in LB, LB supplemented with HPF or 3.5% HSA, were used to determine the MICs of CFDC. Ten strains had baseline resistance to CFDC (Appendix A).

An increase in the MIC values was observed in 14 strains when the cells were growing in the presence of HSA or HPF (Table 1 and Figure 2D). We draw attention to the occurrence of heteroresistant colonies, with the inhibition ellipse, in 11 of the tested strains (Table 1 and Appendix A). Most of the strains that exhibited this phenomenon harbored *bla*_PER-7_ [54]. Our results agree with a recent report indicating that PER-like β-lactamases contribute to a decreased susceptibility to CFDC in *A. baumannii* [45]. In addition, they observed that the combination of CFDC with avibactam may inhibit the activity of PER-type ß-lactamases [45].

## 4. Discussion

Immediately after the invasion of infecting bacteria, the body responds by reducing free iron levels in the blood and tissues (hypoferremic response) [70,71,72]. In the face of these nutritional limitations, microorganisms evolve strategies to scavenge sufficient trace elements necessary to support their metabolism and growth. Among them, siderophore-mediated iron uptake systems are widespread among bacterial pathogens and are the systems utilized by CFDC to increase cell entrance.

Previous reports support a link between mutations in iron transporters and a decrease in susceptibility to CFDC [33,42,44]. In particular, PirA and PiuA have been linked with CFDC entry and CFDC resistance when mutations in these genes have been reported [33,34]. Our results show a decrease in the expression of genes coding for the TonB-dependent receptors BauA, PirA, and PiuA, when CRAB strains are exposed to human fluids and HSA. In addition, the increased expression of β-lactam resistance-associated genes, in particular, the increased expression of genes encoding PBPs in the three tested conditions, was also observed. These two observations are coupled with elevated CFDC MICs in 14 CRAB isolates when exposed to HPF and/or HSA.

Supporting our observations, recent studies have shown that the down-regulation of *pirA* and *piuA* in three *A. baumannii* clinical strains is associated with resistance to CFDC [33,44]. In addition, this work showed that the alteration of the hydrophobic isoleucine to the charged asparagine, at position 236 of PBP3, could also contribute to the CFDC resistance [44].

In other Gram-negatives, the relationship between mutations in TonB-dependent catecholate siderophore receptors and CFDC was observed. A recent case report showed the development of resistance to CFDC in a patient with intra-abdominal and bloodstream infections caused by a CR-*Enterobacter cloacae* strain after 21 days of CFDC treatment. Whole-genome sequencing revealed heterogeneous mutations in the *cirA* gene, which encodes a TonB-dependent catecholate siderophore receptor, conferring phenotypic resistance to cefiderocol. *cirA* was the only catecholate receptor present in that clinical isolate, supporting the role of the mutation of this gene in CFDC resistance [73]. Moreover, in *Pseudomonas aeruginosa*, mutations in *piuD* and *pirR*, major iron transport pathways, and a leucine-to-phenylalanine substitution at amino acid position 147 in the ADC β-lactamase, were associated with CFDC MIC increases [73,74]. Simner et al. reported that substitutions in the region of the AmpC omega loop contribute to increased cefiderocol MICs to *P. aeruginosa* [75].

The present work focuses on the effects of different human bodily fluids on iron uptake and antibiotic resistance genes, some of which were linked with the reduced susceptibility to CFDC. Our results show a down-regulation or up-regulation of the genes associated with iron uptake or β-lactam resistance. These alterations were seen among the different strains and under different conditions. We believe that the sum of all the changes may contribute to the reduced effectiveness of CFDC when studied in vitro.

The exposure of *A. baumannii* to human fluids results in the transcriptional repression of genes coding for proteins involved in TonB-dependent siderophore-mediated iron acquisition. Taking into account these observations and the fact that HPF and HS possess a high concentration of HSA, a protein that has been reported as an important host iron reservoir [76], it is possible to speculate that the reduced efficacy of CFDC in the presence of these fluids is a consequence of the iron-rich environment created by the presence of HSA. In the case of HPF, we cannot discount the possibility that other proteins such as ferritin, which are abundant in HPF, may provide the necessary metal ions through this source, reducing the expression of the catechol iron uptake system and the number of entry sites for cefiderocol, leading to diminished efficacy.

## 5. Conclusions

Overall, our results show that *A. baumannii* may be unique in its response to human bodily fluids by modulating the expression of iron uptake genes and β-lactam resistance-associated determinants. We hypothesize that the presence of iron-binding proteins in the human fluids is sensed by *A. baumannii,* which in turn responds by down-regulating the expression of iron uptake system genes, impacting CDFC activity. The particular changes in expression of the aforementioned group of genes, which is not evaluated in traditional susceptibility tests, may contribute to increases in MICs. These results also raise questions regarding the expression of other factors that may contribute to challenges in overcoming infections by *A. baumannii.*

The limitations inherent in performing MICs in the presence of bodily fluids (protein binding) are recognized. We are also aware that comparing these analyses does not take into account the pharmacokinetic and pharmacodynamic (PK/PD) parameters that ensure drug efficacy [36,77]. Lastly, we appreciate that these data must also be evaluated in the context of outcomes in animal models of infection [78,79,80]. Nevertheless, the impact on iron transport mechanisms uncovered in these observations merits further analyses.

## Figures and Tables

**Figure 1 biomedicines-10-00600-f001:**
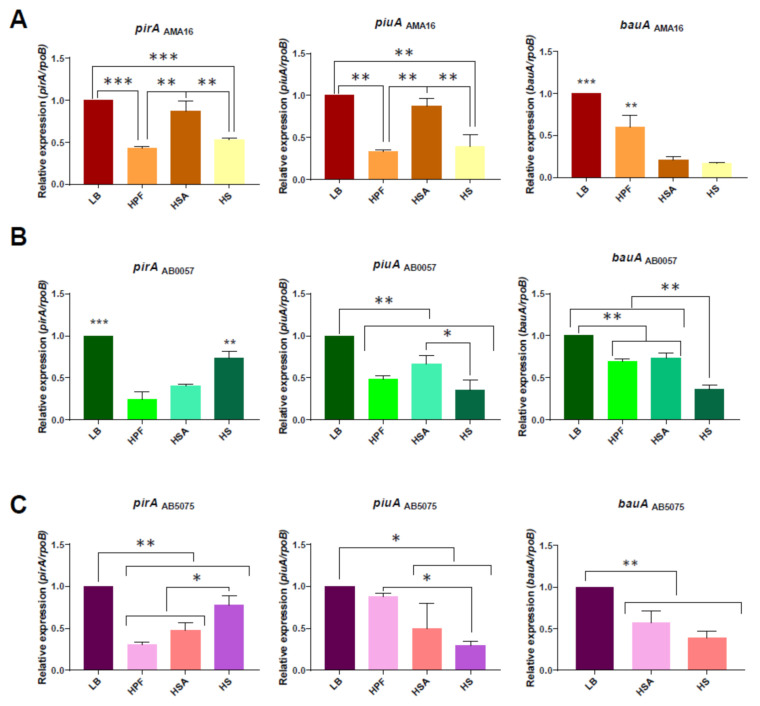
Genetic analysis of iron uptake genes of AMA16 (**A**), AB0057 (**B**), and AB5075 (**C**) *A. baumannii* strains and qRT-PCR of genes associated with iron uptake, *pirA, piuA,* and *bauA,* expressed in LB, LB supplemented with HPF or with HSA, or cultured in HS. Fold changes were calculated using double ΔCt analysis. At least three independent samples were used, and four technical replicates were performed from each sample. The LB was used as reference. Statistical significance (*p* < 0.05) was determined by ANOVA followed by Tukey’s multiple comparison test: one asterisk: *p* < 0.05; two asterisks: *p* < 0.01; and three asterisks: *p* < 0.001.

**Figure 2 biomedicines-10-00600-f002:**
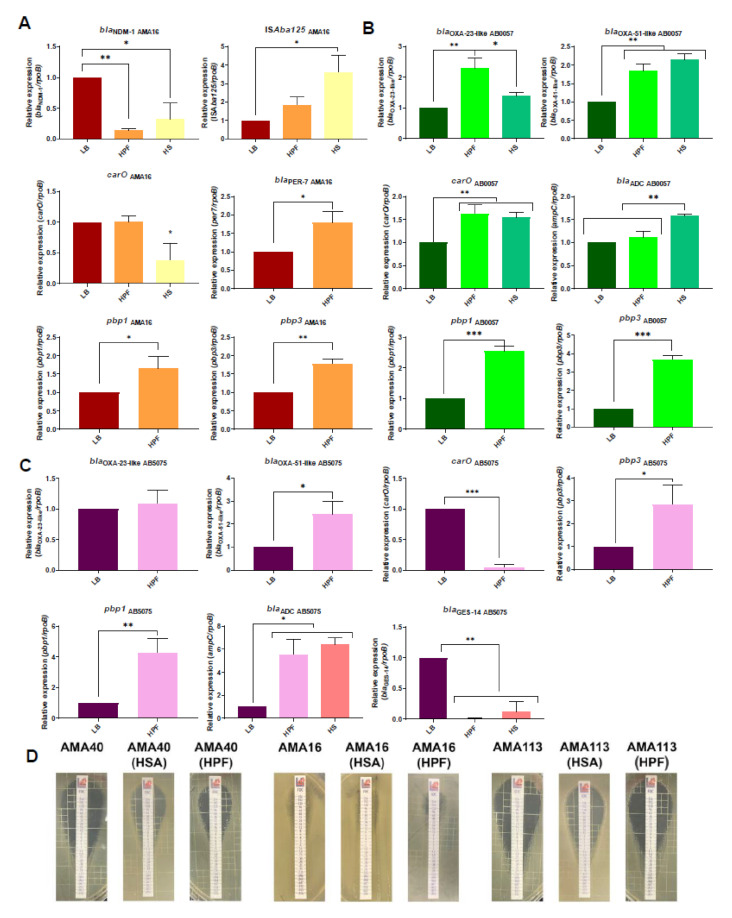
(**A**–**C**) Genetic analysis of β-lactamase and PBP genes of AMA16 (**A**), AB0057 (**B**), and AB5075 (**C**) *A. baumannii* strains and qRT-PCR of genes associated with β-lactams resistance expressed in LB, LB supplemented with HPF, or in HS. Fold changes were calculated using double ΔCt analysis. At least three independent samples were used. LB was used as the reference condition. Statistical significance (*p* < 0.05) was determined by ANOVA followed by Tukey’s multiple comparison test: one asterisk: *p* < 0.05; two asterisks: *p* < 0.01; and three asterisks: *p* < 0.001. (**D**) Effect of HSA and HPF on the antimicrobial susceptibility of *A. baumannii* representative strains AMA40 and AMA113 grown in LB broth, LB broth plus 3.5% HSA, or HPF, that were used to perform cefiderocol (CFDC) susceptibility. Minimum inhibitory concentration (MIC) on cation-adjusted Mueller–Hinton agar was performed by MTS (Liofilchem S.r.l., 64026 Roseto degli Abruzzi TE, Italy), following the manufacturer’s recommendations.

**Table 1 biomedicines-10-00600-t001:** Minimal inhibitory concentrations of cefiderocol (CFDC) for 22 carbapenem-resistant *Acinetobacter baumannii* representative strains performed using CFDC MTS strips (Liofilchem S.r.l., 64026 Roseto degli Abruzzi TE, Italy) on Mueller–Hinton agar (cation adjusted). *A. baumannii* cells were cultured in LB or LB supplemented with 3.5% HSA or HPF, respectively.

CFDC MICs (mg/L)			
Strain	LB	HPF	3.5% HSA
AB5075	0.5 (S)	1 (S)	2 (S)
ABUH702	0.38 (S)	1.5 (S)	3 (S)
AMA16	>4.5 * (I)	>256 (R)	32 * (R)
AB0057	1 (S)	8 (I)	1.5 (S)
AMA40	0.5 (S)	16 * (R)	3 (S)
AMA41	0.094 (S)	0.5–0.75 (S)	2 (S)
AMA113	0.5 (S)	1.5 (S)	1.5 (S)
AMA181	0.19 (S)	0.19 (S)	0.75 (S)
AMA3	24 (R)	>256 (R)	32 * (R)
AMA4	16 * (R)	48 * (R)	64 * (R)
AMA5	>256 (R)	>256 (R)	16 * (R)
AMA9	32 (R)	48 (R)	16 (R)
AMA14	8 * (I)	16 * (R)	12 (I)
AMA17	>256 (R)	>256 (R)	>256 (R)
AMA18	64 * (R)	16 * (R)	16 * (R)
AMA19	4 (S)	4 (S)	48 * (R)
AMA28	32 * (R)	>256 (R)	32 * (R)
AMA30	64 * (R)	128 * (R)	12 * (I)
AMA31	>256 (R)	>256 (R)	96 * (R)
AMA33	16 * (R)	>256 (R)	>256 (R)

* Intra-colonies are present. S: Susceptible, I: Intermediate, R: Resistant.

## Data Availability

All data pertaining to the study described in the manuscript is described in the report.

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
