# Peer review of "Human Serum Proteins and Susceptibility of Acinetobacter baumannii to Cefiderocol: Role of Iron Transport"

_biomedicines, 2022, doi:10.3390/biomedicines10030600_

Round 1

Reviewer 1 Report

The authors presented the paper "Human Serum Proteins and Susceptibility of Acinetobacter baumannii to Cefiderocol: role of iron transport".

It is known that HSA can bind RNA and DNA and has RNA hydrolyzing activity. In this way your experiment in par 2.2. may lead to not all RNA after incubation with such amount of albumin. That is why some of your effects may have other explanations. Have you done some control experiments? Maybe your effect is much higher or lower.

https://www.sciencedirect.com/science/article/abs/pii/S0960894X08011086?via%3Dihub

Author Response

Author’s Response:

We appreciated the reviewers’ comment.

If we understand correctly, the reviewer’s argument, i.e., a reduction in RNA when the cells were cultured in the presence of HSA may be due to the HSA’s RNA-hydrolyzing activity, requires that HSA contacts the RNA that will be hydrolyzed. This is not the case; HSA and the other fluids tested as inductors were added to the growth medium (LB) medium, and the cells were allowed to grow in these conditions for 5 h at 37ºC. After that, the cells were collected, and RNA was extracted. Since HSA does not reach the cytosol, at least not in an intact manner, hydrolysis seems not possible during cell growth. It could be argued that some RNA is hydrolyzed during extraction, but for this to be the case, one should assume that the cells were not well washed after being pelleted by centrifugation, which is not the case. Therefore, the RNA is intracellular at all moments during the experiment, and the HSA stays in the milieu.

It is also worth mentioning that the presence of HSA in the growth medium produced a reduction only in some mRNA species. These results speak against an unspecific hydrolyzing effect.

We also want to underscore that all RNA samples were checked after extraction. Agarose gels electrophoresis confirmed the integrity of RNA and lack of DNA contamination. The same cDNA amount for every condition was used to perform qRT-PCR assays.

Finally, the article provided by the reviewer describes HSA hydrolysis of extracellular, not intracellular, RNA. 

Reviewer 2 Report

The authors extensively study the effects of human serum, human serum albumin, or human pleural fluid  on Cefiderocol sensitivity of a panel of A. baumanii strains. Cefiderocol is a cidal antibiotic. It would be of great interest if for the key results the authors would also provide MBC and / or time-resolved kill-curves. I believe this kind of data would be of great relevance for assessing the clinical relevance of the observed effects.

Author Response

Author’s Response:

We appreciate the reviewers’ comment and the suggestion.

We agree that confirming that the effect of cefiderocol is bactericidal under the conditions tested is important. We have indeed carried out some experiments in the presence of human fluids. They actually show that adding these components to the growth medium does not modify the bactericidal nature of the action of the antibiotic.

A follow-up project that will be submitted for publication in the near future extends the studies of the effects produced by human fluids, and their components will include the experiments demonstrating a bactericidal effect of cefiderocol. We would prefer to include those results in the upcoming publication.